# The Imitation of the Ovarian Fatty Acid Profile of Superfertile Dummerstorf Mouse Lines during IVM of Control Line Oocytes Could Influence Their Maturation Rates

**DOI:** 10.3390/biomedicines11051439

**Published:** 2023-05-13

**Authors:** Michela Calanni-Pileri, Marten Michaelis, Martina Langhammer, Paolo Rosellini Tognetti, Joachim M. Weitzel

**Affiliations:** 1Institute of Reproductive Biology, Research Institute for Farm Animal Biology (FBN), 18196 Dummerstorf, Germany; calanni-pileri@fbn-dummerstorf.de (M.C.-P.); michaelis@fbn-dummerstorf.de (M.M.); rosellini-tognetti@fbn-dummerstorf.de (P.R.T.); 2Service Group Lab Animal Facility, Institute of Genetics and Biometry, Research Institute for Farm Animal Biology (FBN), 18196 Dummerstorf, Germany; martina.langhammer@fbn-dummerstorf.de

**Keywords:** high-fertility phenotype, outbred mouse line, biodiversity, fatty acids during IVM, optimization of the IVM system

## Abstract

Declining human fertility worldwide is an attractive research target for the search for “high fertility” genes and pathways to counteract this problem. To study these genes and pathways for high fertility, the superfertile Dummerstorf mouse lines FL1 and FL2 are two unique model organisms representing an improved fertility phenotype. A direct reason for this remarkable characteristic of increased litter size, which reaches >20 pups/litter in both FLs, is the raised ovulation rate by approximately 100%, representing an impressive record in this field. Dummerstorf high-fertility lines incarnate extraordinary and singular models of high-fertility for other species, mostly farm animals, with the aim of improving production and reducing costs. Our main goal is to describe the genetic and molecular pathways to reach their phenotypical excellence, and to reproduce them using the control population. The large litter size and ovulation rate in Dummerstorf lines are mostly due to an increase in the quality of their oocytes, which receive a different intake of fat and are composed of different types and concentrations of fatty acids. As the follicular microenvironment plays a fundamental role during the oocytes development, in the present manuscript, we tried to improve the in vitro maturation technique by mimicking the fatty acid profile of FLs oocytes during the IVM of control oocytes. Currently, the optimization of the IVM system is fundamental mostly for prepubertal girls and oncological patients whose main source of gametes to restore fertility may be their maturation in vitro. Our data suggest that the specific fatty acid composition of FLs COCs can contribute to their high-fertility phenotype. Indeed, COCs from the control line matured in IVM-medium supplemented with C14:0 (high in FL2 COCs) or with C20:0, C21:0, C22:0, and C23:0 (high in FL1 COCs), but also control oocytes without cumulus, whose concentration in long-chain FAs are “naturally” higher, showing a slightly higher maturation rate. These findings represent an important starting point for the optimization of the IVM system using FA supplementation.

## 1. Introduction

### 1.1. IVM (In Vitro Maturation)

One of the biggest scientific issues of the 21st century is the decline in human fertility worldwide, which is mostly due to social and economic reasons [1]. The main consequence of this decrement is the enlarging application, by couples experiencing trouble with producing babies, of in vitro fertilization cycles, which in turn are obviously prone to select the same “low-fertility” genes already present in the parental genome [1]. In addition, those patients must undergo hormonal treatments to obtain as many mature oocytes as possible for the next fertilization step. Moreover, prepubertal girls with cancer or at high risk of sterility are incapable of using IVF cycles to save embryos for later use [2]. Usually, in young patients as well as in oncological women, the most commonly used technique to conserve fertility is the preservation of ovarian tissue before starting chemotherapy and reimplantation after recovery, with the risk of implanting new metastases together with the tissue [2]. To avoid all of these problems, scientists are struggling with finding the perfect culture system for maturing immature oocytes in vitro and maximizing egg yield [2,3,4,5] which can also be of help in women whose infertility is caused by polycystic ovarian syndrome (PCOS). Approximately one million oocytes present at birth die by atresia, and only a few hundred can be ovulated. The ability to rescue these oocytes destined to die and mature them in vitro would provide an invaluable source of oocytes for infertile women [6]. Obviously, a good follicular environment is of primary importance for producing developmentally competent and healthy oocytes [6]. In this context, it is necessary to discover and study models of increased fertility traits, which can make a difference in optimizing maturation/fertilization culture systems and mimicking their way of managing.

### 1.2. Superfertile Mouse Lines

The two Dummerstorf high fertility mouse lines FL1 and FL2 have been bred for the phenotype “high fertility” for >50 years [7]. Simultaneously, with FLs, a control line derived from the same founder population and genetic background has been bred without any selection trait (Dummerstorf FZTDU, control). In >200 generations, FLs doubled the number of offspring per litter compared to unselected control mice from approximately 11 to >20 and >21, respectively, for FL1 and FL2, and one of them (FL1) can deliver up to nine litters (approximately 86 pups) per lifetime [8]. Unlike the other animal models, which are mostly transgenic or knockout and where the single alteration is known and it is easier to find the pathways to the related phenotype, Dummerstorf models are instead selected for specific wished characteristics but with the cardinal aim of discovering the corridor of unknown molecular/genetic mechanisms behind them. In the field of reproductive biology, our Dummerstorf lines represent two of the <1% of models of increased fertility. Two other examples are the Gpr149-knockout females [9] and transgenic mice overexpressing Bcl2 [10], where the litter size increased only by 10–20%, which is very far from the 90–95% increase obtained by FL1 and FL2 selected lines. The main reason is to be found in the way in which the lines were obtained, and therefore in the substantial difference between single-gene and whole genome alterations. We assume that FLs are useful models for farm animal/livestock studies to improve production and sustainability and reduce economic and natural resources. We recently found that the main reason for the increased ovulation rate is the ability of fertility lines to increase the quality of the oocytes per ovary compared to control females for comparable quantities [11]. Moreover, the composition and distribution of adipose tissue in high-fertility females seemed to support the hypothesis according to which a higher quantity of visceral or ovarian fat with a certain composition can improve their oocyte capacity [12]. In addition, by analyzing the fatty acid profile of several tissues in the fertility lines and the control population, we found large differences between the lines not only in the plasma, liver, and adipose tissue but also within the gonads, with the FL1 line carrying the largest differences [12].

These findings represent the starting gate of the present work. Indeed, we extended our previous data [12] using a more profound interpretation, combining and comparing the variations between FA composition in FLs cumulus-oocyte-complexes (COCs), granulosa cells/cumulus cells, and oocytes without cumulus cells (WOCs) as well as the main differences with the control population in the same samples. More precisely, we focused only on the FAs with higher concentrations in COCs and lower concentrations in granulosa cells in fertility lines, which showed an opposite status in control females. In addition, we tested those particular FAs during an IVM (in vitro maturation) experiment of immature control oocytes (germinal vesicles) and calculated the maturation rate in comparison with control oocytes matured in an IVM-medium without FA supplementation. Currently, fatty acids are studied worldwide in maturation and fertilization experiments. Nevertheless, the literature mostly describes saturated fatty acids (SFAs) as worse than mono- and poly-unsaturated FAs (MUFAs and PUFAs) for oocyte/embryo development [13,14,15] but better for granulosa cells culture [16]. In contrast, in our study we found that FL1 showed increased levels of long-chain SFAs C20:0, C21:0, C22:0, and C23:0 in COCs, lowering their presence in granulosa cells, and the opposite was found in control females, which led us to hypothesize that within the COCs, those FAs are situated inside the oocyte and not in the cumulus cells around it. We speculate that some long-chain SFAs can improve the maturation rate of control COCs if added to the IVM-medium, following the “way of managing” of FL1 COCs. In particular, we observed that one particular fatty acid, C23:0, was “naturally” more concentrated in the WOCs of control mice than in the WOCs of all the other lines, as well as more than the COCs and granulosa cells of the same line. Indeed, comparing the maturation rate of WOCs and COCs of control females (without FA supplementation), we obtained an increased maturation rate in WOCs that confirms our first hypothesis. We would like to continue the research on IVM –IVF of murine oocytes and other species using long-chain SFAs C20:0, C21:0, C22:0, and C23:0 to demonstrate their ability to increase the maturation rate and consequently the fertilization rate during assisted reproduction treatments. In addition, it would be a direct consequence to test the administration of the same FAs in vivo, in control mice first and other murine models and different species later to measure if they can have an influence in increasing the ovulation rate of those animals and raise the maturation rate. In recent years, Dummerstorf superfertile lines have shown some unusual characteristics for models of higher fertility, for example, an elongated estrous cycle and atypical levels of insulin/leptin and glucagon depending on the line [17], together with an increased quantity of adipose tissue [12] that are usually described in the literature as correlated with lower fertility or infertility. In those lines, abdominal fat and adipose tissue around the ovary have a different composition and indeed contain a different quantity of lipids, which is higher in the ovary than in the abdomen and can be helpful for influencing the quality of the ovarian environment and consequently, the oocyte quality, thus intensifying the ovulation rate [12]. Given their qualities and curious/peculiar characteristics, these lines could be useful not only for the study of other species but also for those showing the same “fertility issues” that can be reversed using FLs’ problem-solving skills.

## 2. Results

### 2.1. Fatty Acid Concentration in COCs-WOCs-Granulosa Cells

We unfolded our previous data (raw data from Supplementary Material of Calanni-Pileri et al., Sep 2022, plus partially on Figure 10 of the same publication [12]) concerning the comparison between COCs, WOCs, and granulosa cells, extending them to a deep analysis of the composition of SFAs C14:0, C16:0, C20:0, C21:0, C22.0, and C23:0 in fertility lines and control females (Figure 1). We found that these fatty acids were more concentrated in FLs COCs (particularly C14:0 and C16:0 in FL2 and C20:0, C21:0, C22.0, and C23:0 in FL1) than in control COCs. Moreover, they showed a lower percentage in FLs granulosa cells compared to control granulosa cells with the exception of C14:0. Specifically, this is in regard to the data concerning long-chain saturated fatty acids.

C20:0, C21:0, C22:0, and C23:0 were statistically significant (Figure 1). In addition, we found a difference between the levels of C23:0 in the WOCs that were significantly higher in the control line than in the FLs. For this reason, we checked the differences between WOCs and COCs concerning the complete fatty acid panel by the PCA and heatmap analysis to visualize the largest differences between the lines and types of oocytes (Supplementary Materials of Calanni-Pileri et al. Sep 2022 [12], and Figure 2 of the present manuscript). Based on their FA profiles (Figure 2), WOCs and COCs are different families of oocytes, as shown by the PCA (Figure 2A, circles represent COCs and squares represent WOCs). In addition, within the heatmap (Figure 2B), the lines segregate first on the type of oocyte (we see that the WOCs of the three lines are more similar to each other compared to the COCs of the three lines), and then COCs from individuals of the same line are mostly segregated together. From the first rows of the heatmap in Figure 2B (gray arrows), it is easy to understand how the most abundant FAs in the COCs of the FL1 line are SFA C20:0, C21:0, C22:0, C23:0, C24:0, and C26:0 (the pink represents the COCs and the blue represents the FL1 line). From the same rows, we perceived that the same fatty acids in WOCs (blue/green AND purple) have lower concentrations in fertility lines compared to the FLs’ COCs (blue/green AND pink) but also compared to the control WOCs (red and purple).

### 2.2. Maturation Rates

We performed the IVM exclusively using oocytes of the control line. Similar rates of maturation were found in nonsupplemented COCs and COCs supplemented with C16:0. COCs matured in an IVM-medium supplemented with C14:0 or the mixture of C20:0, C21:0, C22:0, and C23:0 showed a tendency to increase the maturation rates (Figure 3A,B, not significant). As WOCs of the control line contain higher levels of C23:0 and C22:0 compared to the COCs of the same line, to demonstrate our hypothesis, we performed an additional IVM experiment using only COCs and WOCs of the control line, both nonsupplemented, to measure whether control WOCs can mature better than control COCs. From the IVM experiment, we obtained a higher percentage of maturation in oocytes without cumulus compared to the COCs, without the supplementation of any fatty acid, which confirms our hypothesis as the WOCs “naturally” contain a higher concentration of those FAs that help the FL1 COCs to grow better (Figure 3). In addition, we compared the nonsupplemented-control WOCs with control-WOCs supplemented them with the fatty acids that gave us the highest maturation rates, C14:0 (50%), and the mixture of C20:0, C21:0, C22:0, and C23:0 (12.5% each). We obtained a higher percentage of mature oocytes without cumulus that was nonsupplemented (Figure 3), which is additional proof that the FAs already present in the WOCs are sufficient to obtain a good level of maturation. In addition, as we obtained higher maturation rates in the supplemented WOCs than in the COCs (both concerning the nonsupplemented and supplemented ones), we speculate that either 50 µM is not enough for a good level of maturation in COCs or this is due to the chosen combination of FAs.

## 3. Discussion

In the field of reproductive medicine, the practice of in vitro culturing of immature oocytes is advancing rapidly. The main reason is its unique advantage of being able to obtain a large number of embryos, avoiding both the hormonal stimulation in women and reimplantation of ovarian tissue after chemotherapy treatments in ex-oncological patients. In addition, this clinical practice would be of great relevance in prepubertal girls in need of ovarian removal [19,20,21]. The aim of IVM is to create a perfect environment with the best culture conditions to ensure the development of healthy oocytes [21,22]. In recent decades, many improvements have been made to increase the number of embryos and offspring starting with oocytes grown in vitro [4]. In farm animals (pig, cattle, goat), IVM has been successful [23,24,25,26]. However, the success rate of embryo production from in vitro matured oocytes, mostly in humans but also in animals, is still lower than that from in vivo matured oocytes obtained after hormonal stimulation [27,28].

In this scenario, the fertility traits of Dummerstorf superfertile mouse lines, mostly FL1 but also FL2, can represent a great model for achieving the objective of improving IVM culture conditions following the molecular skills of FLs. Dummerstorf high-fertility lines have been selected for more than 50 years following the selection trait of larger litter size [7]. FLs obtained a 100% increase in the number of pups per litter compared to an unselected control line from the same founder population by increasing their ovulation rate by >100% [29,30]. In turn, the larger number of ovulated oocytes per cycle is due to a gain in the quality of their oocytes and follicles compared to the control [11]. We previously described that the enhanced number of high-quality oocytes in fertility lines could be an effect of the characteristic microenvironment of the ovary, and in particular, the quantity and location of adipose tissue around the ovary as well as the form of fatty acids and their distribution inside the follicles and the oocytes [12]. As an example, our superfertile mice are characterized by an increased quantity of fat in the abdomen as well as outside the ovary, which can allow their oocytes to incorporate more energy and grow better [12]. In addition, the fatty acid composition of the adipose tissue around the ovary as well as in their follicles and oocytes is different from that of the unselected control line, mostly in FL1. At the molecular level, FL1 mice contain higher amounts of MUFAs in adipose tissue and liver but have low concentrations of PUFAs in adipose tissue [12]. Both fertility lines show increased concentrations of unsaturated fatty acids and, in contrast, decreased concentrations of saturated fatty acids in granulosa cells [12]. With the present manuscript, we wanted to draw attention to the fatty acid content of fertility lines oocytes and to recreate the adipose microenvironment of FLs’ ovaries in the control line using cumulus–oocyte complexes of the control population to determine whether a higher amount of specific fatty acids during in vitro maturation of immature COCs could increase the maturation rate. Aizawa et al. recently described how the presence of lipids is of primary importance during in vitro development, and that even when eliminating all the lipids from the oocytes, the cells recreate them right away [31].

We strictly extended the fatty acid composition analysis of the COCs, WOCs, and granulosa cells of our three mouse lines (control, FL1, FL2) and observed that the only FAs that were more concentrated in the FLs’ COCs vs. the control and contemporaneously less concentrated in FLs granulosa cells vs. the control were C16:0 (in FL2), C20:0, C21:0, C22:0, and C23:0 (in FL1). In addition, we noticed an increase in C14:0 in FL2 COCs compared to the control, even with no difference in granulosa cells (Figure 1). 

We chose a concentration of 50 µM based on the findings of our last publication (Calanni-Pileri et al. Sep. 2022, Supplementary Materials [12]) where FLs showed a slight increase in COCs compared to control in C14:0 and C16:0 (in FL2) of approximately 5% each, and of C20:0, C21:0, C22:0, and C23:0 (in FL1) of approximately 5% in total, which is the reason why we used them together. Even if the concentration usually used in the literature is higher [15,32,33], as the difference between the lines was only moderate, we decided to use 50 µM as fatty acids that are already present in control oocytes and our aim was to increase their concentration to reach that found in the FLs.

We obtained more mature oocytes in the presence of very-long-chain fatty acids (C20:0, C21:0, C22:0, and C23:0) and/or in the presence of C14:0. Those results, even though they were not significantly higher, represent a very good starting point for implementing the IVM method not only in mice but also in farm animals and human fertility care. For an even better demonstration of the “helper” competencies of these fatty acids, we stepped forward with another experiment using control oocytes again, but this time WOCs and COCs were nonsupplemented. 

We observed that in only the control line, the oocytes without cumulus have naturally higher amounts of some long-chain fatty acids, and between them, there are the SFAs C22:0 and C23:0 ([12] and Figure 2). For this reason, we compared the maturation rate between COCs and WOCs in the control line without the supplementation of any FA to demonstrate that the WOCs can mature as much as the COCs, or even better because those particular fatty acids can help the process of maturation. Again, we obtain a higher maturation rate in the WOCs compared to the COCs, which demonstrates our hypothesis. In addition, we examined the maturation rate of the WOCs with and without fatty acid supplementation. This time, we chose to add the FAs that worked better in the previous experiment: C14:0 (50%) and the mixture of C20:0, C21:0, C22:0, and C23:0 (50%) to determine whether we obtain even better results if we combine FL1′s with FL2′s profile. From the last experiment, we obtained a higher maturation rate in the nonsupplemented WOCs, which could be due to the concentration of fatty acids (which was maybe too high, as they are already present inside the WOCs) or to the combination (which we would like to further study). Nevertheless, the maturation rates of supplemented WOCs remained higher compared to the supplemented COCs, which could be explained by assuming that cumulus cells absorb the FAs present in the medium, even when they may have a positive impact on the oocyte. We aim to step forward with our studies analyzing IVM using oocytes from different mouse lines and supplementing different concentrations of C14:0, C20:0, C21:0, C22:0, and C23:0 as well as C24:0 and C26:0 [12], alone and in combination, to optimize our IVM method.

## 4. Materials and Methods

### 4.1. Fatty Acids and Media Preparation

Powdered tetradecanoic acid (C 14:0, Matreya LLC, lot n. 1010, State College, PA, USA), palmitic acid (C 16:0, Sigma-Aldrich, lot n. P0500, St. Louis, MO, USA), eicosanoic acid (C 20:0, Matreya LLC, lot n. 1030, State College, PA, USA), henicosanoic acid (C 21:0, Matreya LLC, lot n. 1241, State College, PA, USA), docosanoic acid (C 22:0, Matreya LLC, lot n. 1035, State College, PA, USA), and tricosanoic acid (C23:0, Matreya LLC, lot n. 1186, State College, PA, USA) were first diluted in ethanol (Rotipuren UN1170, art n. 9065 4) to create 12 mM stock solutions. The FAs were then used at a concentration of 50 µM (2 µL in 500 µL of IVM-medium) to obtain an adequate concentration of ethanol (0.4%) that was nontoxic to the oocytes. The media used in the present study were the M7167 M2 medium (Sigma-AldrichSLCF4312 with Hepes and red phenol, without penicillin and streptomycin, St. Louis, MO, USA) and handmade IVM-medium. The IVM-medium was prepared using a tissue culture medium (TCM-199) as a base medium supplemented with 25 mM HEPES, 0.22 mg/mL sodium pyruvate, 200 µg/mL gentamicin, 50 ng/mL IGF, 10 ng/mL EGF, 10% V/V PFF (porcine follicular fluid), and 10% *v*/*v* FBS as per a previously described method [34]. This handmade IVM-medium has been chosen because it provided the best results in terms of maturation even in our unselected control population.

### 4.2. Mice

The animal experiments were performed following European legislation on laboratory animals care and use and the guidelines of local authorities (Land Mecklenburg-Vorpommern, Germany). Starting in the 1970s, scientists at the Research Institute for Farm Animal Biology (FBN Dummerstorf, Germany) began to cross four outbred (NMRI origin, Han: NMRI, Han: CFW, Han: CF1) and four inbred (CBA/Bln, AB/Bln, C57BL/Bln, XVII/Bln) mouse lines to obtain a heterogeneous founder population. This founder population was maintained without any selection for 200 generations and 50 years at a population size of 125–200 breeding pairs per generation. This non-selected control line is called the “ctrl” or “control line” and has a normal litter size of about 11.5 pups per litter. From the same founder animals (in the 1970s), two lines were created using the fertility index: 1.6 × litter size + litter weight of total pups [7]. After 200 generations and 50 years of selection with a population size of 60–100 breeding pairs, these two fertility lines (FLs; FL1 and FL2) doubled in litter size to 20.6 and 21.4 pups per litter, respectively, with no evidence of growth retardation in a single offspring [8]. Sixty-two female animals belonging to the unselected control line (gen. 207) were housed in a 12–12 h light-dark cycle (6 a.m–6 p.m) with a constant temperature of 22.5 °C and food (ssniff M-Z autoclavable, Soest, Germany) and water provided ad libitum under SPF (specified pathogen free) conditions. Estrous stage identification was performed only by observation of the vaginal opening [35], and mice (17–18 weeks old) with pale vaginal tissues that were completely dry and closed (not pink, not swollen, and not moist) were sacrificed by cervical dislocation. The estrous cycle was evaluated again by vaginal smear postmortem, and mice in the diestrus stage (thirty-nine in total) [17] were then dissected. Two of them were used for the initial training and to stabilize the protocol.

### 4.3. Sample Collection and In Vitro Maturation (IVM)

Both ovaries were extracted and placed into a plate with a prewarmed M2 medium (Figure 4, steps 1 and 2), and then pinned with a 12-needle pin for approximately 2 min. For the assessment of the IVM, cumulus –oocyte complexes (COCs) and oocytes without cumulus (WOCs) fluctuating in the medium were first washed in an IVM-medium and then moved and distributed in six different wells containing IVM-medium supplemented or nonsupplemented with FAs (Figure 4, steps 3 and 4). The final concentration was 50 µM of FAs and 0.4% of ethanol. The total number of oocytes picked up from the thirty-seven mice was 895 COCs and 374 WOCs. The oocytes from three of the thirty-seven mice were not taken into account for the statistical analysis because maturation was assessed only after 16 h. The total number of oocytes used for the statistics was 817 COCs and 359 WOCs:n. 205 COCs—nonsupplemented, ctrln. 203 COCs—with tetradecanoic acid (C 14:0)n. 205 COCs—with palmitic acid (C 16:0)n. 204 COCs—with a mixture of eicosanoic acid (C 20:0), henicosanoic acid (C 21:0), docosanoic acid (C 22:0), and tricosanoic acid (C23:0) in equal quantities (12.5 µM each).n. 155 WOCs—nonsupplementedn. 204 WOCs—with a mixture of tetradecanoic acid (C 14:0, 25 µM), eicosanoic acid (C 20:0, 6.25 µM), henicosanoic acid (C 21:0, 6.25 µM), docosanoic acid (C 22:0, 6.25 µM), and tricosanoic acid (C23:0, 6.25 µM).

The plates were then placed in the incubator for 24 h at 37 °C and 100% humidity in 5% CO_2_, 5% O_2,_ and 90% N_2,_ as suggested by several laboratory manuals on using IVF in mice (The Jackson laboratory, Bar Harbor, Maine, USA, a workshop on assisted reproductive technologies, and “Manipulating the Mouse embryo: a laboratory manual” [36]).

### 4.4. Assessment of Nuclear Maturation

The maturation rates were calculated based on the presence/absence of the polar body (PB). After 24 h maturation, the oocytes with a visible PB were counted, and the maturation rate was measured (Figure 4, step 5). Some of the oocytes were randomly picked and stained for the presence of the polar body. 

### 4.5. Statistical Analysis 

GraphPad Prism 5 software (GraphPad Software, San Diego, CA, USA) was used to analyze data and draw graphs. One-way ANOVA was used for statistical comparison followed by Bonferroni’s posttest. Data are illustrated as the mean ± SEM (standard error of the mean) and designated significant if *p* < 0.05 (different letters correspond to different levels of significance). The heatmap and PCA were created using the Clustvis web tool [18].

## 5. Conclusions

All these results give rise to a new way of viewing in vitro maturation techniques, which can lead to an increase in the mature oocytes used during IVF in farm animals and humans in cases of extraction of GVs (germinal vesicles) during oocyte pick-up from antral follicles after hormonal stimulations. More importantly, IVM can quickly become the easiest way to obtain a large amount of oocytes and embryos, avoiding both hormonal stimulation in women and prepubertal girls, and the reimplantation of ovarian tissue in ex-oncological patients to avoid any relapses. Although the IVM technique is widely studied, the use of FA supplementation during the IVM process still needs to be optimized. However, due to the molecular analysis of our Dummerstorf superfertile mouse lines, we can determine the perfect types and concentrations of the FAs by studying their ovarian microenvironment to obtain quick improvements in this technique and ideally obtain higher maturation rates (in vitro) and ovulation rates (in vivo) by following their “way of managing”. In addition, further progress on the IVM system could be reached using FLs granulosa cells during the IVM of oocytes from different mice/species.

## Figures and Tables

**Figure 1 biomedicines-11-01439-f001:**
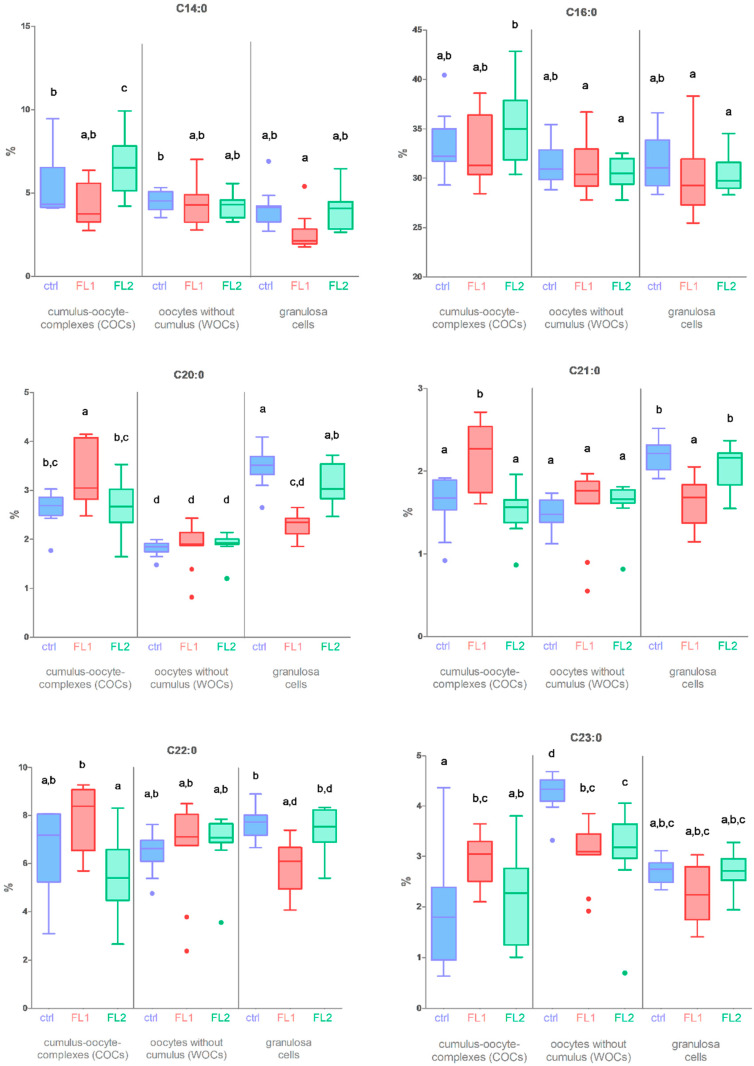
Percentages of fatty acids C14:0, C16:0, C20:0, C21:0, C22:0, and C23:0 in COCs, WOCs, and granulosa cells in control (ctrl, blue), FL1 (red), and FL2 (green). Data were taken from our latest publication (Calanni-Pileri et al. Sep. 2022, Supplementary Materials [12]). GraphPad Prism 5 software (GraphPad Software, San Diego, CA, USA) was used to analyze the data. One-way ANOVA was used for statistical comparison followed by Bonferroni’s posttest. Data are illustrated as the mean ± SEM (standard error of the mean) and designated significant if *p* < 0.05 (different letters correspond to different levels of significance, e.g., “a” and “b”, or “a,b” and “c” are significantly different, while “a” and “a,b”, or “b” and “a,b” are not.).

**Figure 2 biomedicines-11-01439-f002:**
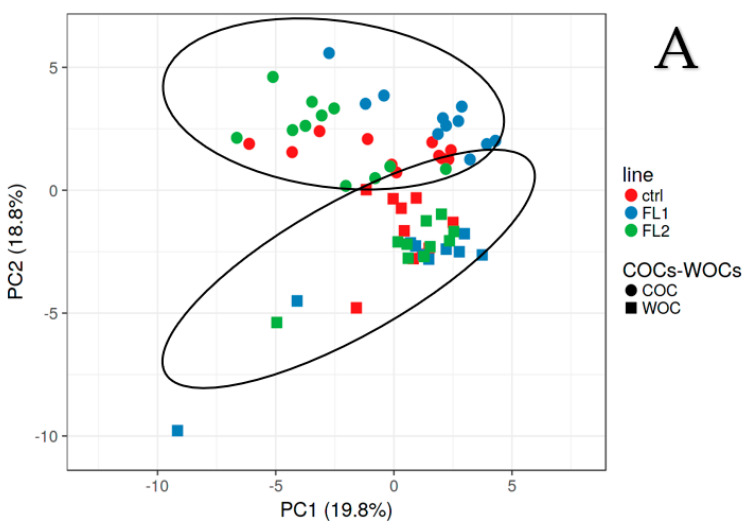
(**A**) Principal component analysis (PCA) and (**B**) heatmap analysis of the fatty acids profile of WOCs and COCs in control (ctrl), as well as the FL1 and FL2 lines. The heatmap and PCA were created using the free Clustvis web tool (http://biit.cs.ut.ee/clustvis/, accessed on 28 December 2022 [18]). In the PCA, the ovals around the circles (COCs) and squares (WOCs) were added to better recognize the two sets.

**Figure 3 biomedicines-11-01439-f003:**
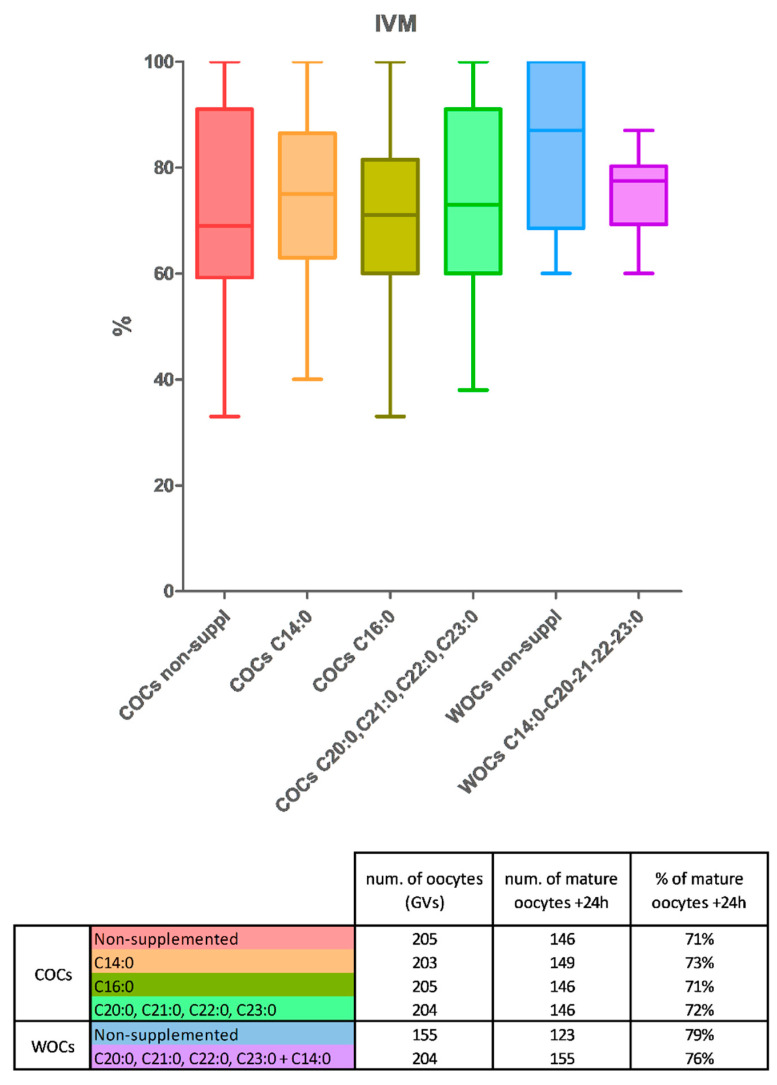
(**Up**) Boxplot representation of in vitro maturation rates of cumulus–oocyte complexes (COCs) and oocytes without cumulus cells (WOCs), nonsupplemented (COCs in red and WOCs in blue), or supplemented with the following fatty acids: C14:0 (orange), C16:0 (olive green), a mixture of C20:0, C21:0, C22:0, C23:0 (25% each) (green), C14:0 (50%), and the mixture of C20:0, C21:0, C22:0, and C23:0 (12.5% each) (purple). GraphPad Prism 5 software (GraphPad Software, San Diego, CA, USA) was used to analyze the data. One-way ANOVA was used for statistical comparison followed by Bonferroni’s posttest. Data are illustrated as the mean ± SEM (standard error of the mean). (**Down**) Table representation of in vitro maturation rates of cumulus–oocyte complexes (COCs) and oocytes without cumulus cells (WOCs), nonsupplemented (COCs in red and WOCs in blue), or supplemented with the following fatty acids: C14:0 (orange), C16:0 (olive green), a mixture of C20:0, C21:0, C22:0, C23:0 (25% each) (green), and C14:0 (50%), and the mixture of C20:0, C21:0, C22:0, and C23:0 (12.5% each) (purple). Data are presented as “total numbers” adding up all the replicates of the same condition. The percentages on column 3 are made by the ratio between columns 1 and 2. To elaborate, e.g., if we have 3/10 mature oocytes on DAY-1, 5/10 on DAY-2, and 2/10 on DAY-3, then for the graph we considered 30% on DAY-1, 50% on DAY-2, and 20% on DAY-3, however in the table we considered the total of 10/30 (33%).

**Figure 4 biomedicines-11-01439-f004:**
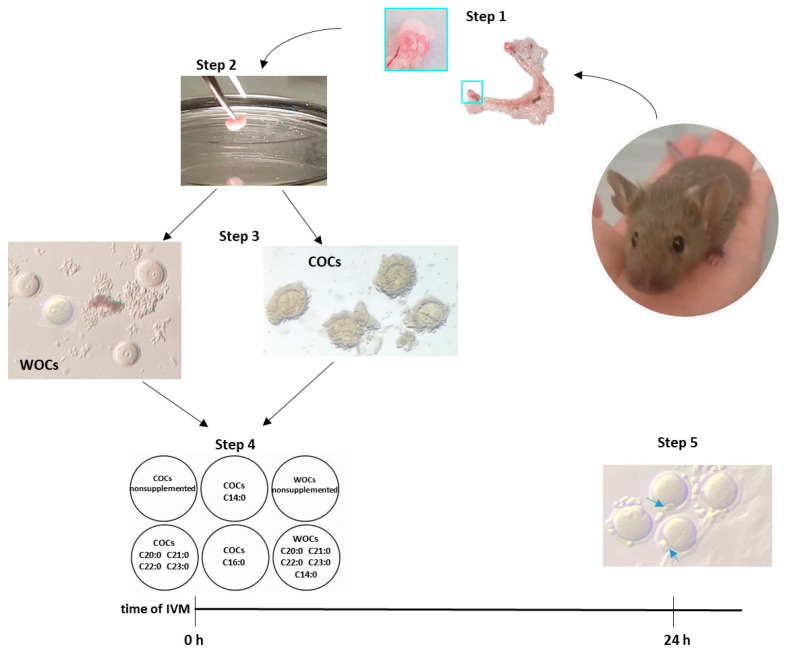
Simple scheme of the material and methods used. Step 1, extraction of the reproductive tract and the ovaries (light blue square). Step 2, pinning of the ovary. Step 3, separation between COCs and oocytes without cumulus (WOCs). Step 4, IVM with or without the addition of fatty acids (depending on the condition). Step 5, assessment of nuclear maturation by the presence of the polar body (blue arrows).

## Data Availability

The data presented in this study are available on request from the corresponding author.

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
