# Peer review of "The Imitation of the Ovarian Fatty Acid Profile of Superfertile Dummerstorf Mouse Lines during IVM of Control Line Oocytes Could Influence Their Maturation Rates"

_biomedicines, 2023, doi:10.3390/biomedicines11051439_

Round 1
Reviewer 1 Report
The article presents quite novel solution that might help to improve GV oocyte maturation in vitro. Any progress in that area is sought and necessary. In my opinion the paper is well designed and well written, to be accepted in the present form.
Author Response
Reviewer: 1
The article presents quite novel solution that might help to improve GV oocyte maturation in vitro. Any progress in that area is sought and necessary. In my opinion the paper is well designed and well written, to be accepted in the present form.
Response: We would like to thank the reviewer for this comment.

Reviewer 2 Report
In the current study, the authors aimed to optimize oocyte IVM protocol by specific fatty acid supplementation on CCs cells lines of Dummerstorf mice. The manuscript is clear and well written. The results are described in details and discussion is in line with the findings.
I just have a minor comment to the authors. During protocol description, they stated they used 50 µM of FAs according to previous results but duration and different combination were not discussed. Did the perform preliminary experiments also on this?
Author Response
Reviewer: 2
In the current study, the authors aimed to optimize oocyte IVM protocol by specific fatty acid supplementation on CCs cells lines of Dummerstorf mice. The manuscript is clear and well written. The results are described in details and discussion is in line with the findings.
I just have a minor comment to the authors. During protocol description, they stated they used 50 µM of FAs according to previous results but duration and different combination were not discussed. Did they perform preliminary experiments also on this?
Response: We would like to thank the reviewer for this comment. We did not test combinations of fatty acids at different concentrations. We checked the concentrations of fatty acids in vivo in our previous work (Calanni-Pileri et al., 2022, Int J Mol Sci; Ref. #1). Based on those concentrations in the oocytes of FLs versus the oocytes in the control line (figure 1 of the present manuscript), we used C14:0 and C16:0 at a concentration of 50 µM and long-chain FAs (C20:0, C21:0, C22:0 and C23:0) at a concentration of 50 µM in total. In preliminary experiments, we analyzed oocyte maturation rates after 12, 16 and 24 hours. We noticed that the oocytes needed 24 hours for good maturation; thus, we focused on this time point.

Reviewer 3 Report
Oocyte maturation in vitro has good advantages for in vitro fertilization. However, there are still many debatable problems applying to infertility treatment. The manuscript “The imitation of the ovarian fatty acid profile of superfertile Dummerstorf mouse lines during IVM of control line oocytes could influence their maturation rates” is devoted to giving information about the effect of fatty acid on oocyte maturation. Also, the authors suggested that the addition of a combination of some specific fatty acids showed an increased maturation rate of oocytes. Although this manuscript has good results, their results need some corrections and additional information.
Especially, it needs additional schemes of their results to make it easier for the reader to read.
Some points have to be corrected.
Major points
1. I understand that Dummerstorf superfertile lines (FL1 and FL2) have higher reproductive performances. However, it needs to add some explanation for the difference between FL1 and FL2. Their’s information in the introduction gives me more confusion than comprehension.
2. What is the difference between the results reported previously and the current results?
3. Are granulosa cells and cumulus cells necessary for the efficient maturation of oocytes?
4. In this study, the authors focused on the selection of fatty acids effective for the maturation of oocytes. It is better to add a schematic model of their relationship and which fatty acid is more effective for WOCs and COCs or granulosa and cumulus cells.
Minor points
1. Line 32: Dose “ctrl” means control?
2. Lin57: Amend “dies” to “die”.
3. Line 102: Add a comma after “Currently”.
4. Line 152: Amend “C22.0” to “C22:0”.
5. In Figure 3, the maturation (%) results in the figure and table are not match.
Author Response
Reviewer: 3
Oocyte maturation in vitro has good advantages for in vitro fertilization. However, there are still many debatable problems applying to infertility treatment. The manuscript “The imitation of the ovarian fatty acid profile of superfertile Dummerstorf mouse lines during IVM of control line oocytes could influence their maturation rates” is devoted to giving information about the effect of fatty acid on oocyte maturation. Also, the authors suggested that the addition of a combination of some specific fatty acids showed an increased maturation rate of oocytes. Although this manuscript has good results, their results need some corrections and additional information.
Especially, it needs additional schemes of their results to make it easier for the reader to read.
Response: We would like to thank the reviewer for this comment. We carefully rephrased several sentences in the Materials and Methods section (lines 325-336), included additional information in figure legend 3 (lines 217-220) and included an additional scheme of the experimental design (novel figure 4, see also below).
Some points have to be corrected.
Major points
- I understand that Dummerstorf superfertile lines (FL1 and FL2) have higher reproductive performances. However, it needs to add some explanation for the difference between FL1 and FL2. Their’s information in the introduction gives me more confusion than comprehension.
Response: Starting in the 1970s, scientists at the Research Institute for Farm Animal Biology (FBN Dummerstorf) began to cross four outbred (NMRI origin, Han: NMRI, Han: CFW, Han: CF1) and four inbred (CBA/Bln, AB/Bln, C57BL/Bln, XVII/Bln) mouse lines to obtain a heterogeneous founder population. This founder population was maintained without any selection for 200 generations and 50 years at a population size of 125-200 breeding pairs per generation. This non-selected control line is called the "ctrl" or "control line" and has a normal litter size of approximately 11.5 pups per litter. From the same founder animals (in the 1970s), two lines were created using the fertility index: 1.6x litter size + litter weight of total pups (Schüler and Bünger, 1982). After 200 generations and 50 years of selection with a population size of 60-100 breeding pairs, these two fertility lines (FLs; FL1 and FL2) almost doubled the litter size to 20.6 and 21.4 pups per litter, respectively, with no evidence of growth retardation in a single offspring (Langhammer et al., 2021). FL1 and FL2 were developed as two independent lines, as FL2 was cycle-synchronized (using chlormadinone acetate) within the first 23 generations. After this initial differential treatment, both lines were maintained independently and developed two distinct physiological, molecular, and endocrine strategies to maintain the high fertility phenotype. We included this information in the Materials and Methods section (lines 325-336).
In the present study, we examined control oocytes and tested whether maturation was enhanced by incubation of these oocytes with the fatty acids previously described to be increased in FL1 and FL2 oocytes.
- What is the difference between the results reported previously and the current results?
Response: In our previous study (Calanni-Pileri et al., 2022, Int J Mol Sci, Ref. #1), we identified different fatty acid compositions in various tissues of the fertility lines FL1 and FL2 compared to the control line in vivo. In Figure 1 of the present manuscript, we analyzed data from COC and granulosa cells that have been shown as raw data in the supplement of Ref. #1. Figure 2 represents a novel analysis of these data. In Figure 3, we used those fatty acids that have previously been found to be increased in FL oocytes cells to test whether an increased concentration of these fatty acids has a beneficial effect on the maturation rates of control oocytes in vitro. This an entirely novel experiment.
- Are granulosa cells and cumulus cells necessary for the efficient maturation of oocytes?
Response: The reviewer is correct that cumulus and/or granulosa cells are very important for proper oocyte maturation in vivo, as has been shown in numerous publications (Arcos et al., 2017, Porras-Gòmez and Moreno-Mendoza, 2020). However, in the present study we wanted to test the influence of different fatty acids on oocyte maturation in vitro +/- granulosa cells. Since we previously described altered fatty acid concentrations in granulosa cells of FL1 and FL2 compared to control line, we wanted to omit these cells in the present experiment (WOC samples, see Figure 3). The WOC samples were analyzed to avoid a mixed effect by both the different fatty acid concentrations of different granulosa cell populations from different mouse line origins plus our external fatty acid treatment.
- In this study, the authors focused on the selection of fatty acids effective for the maturation of oocytes. It is better to add a schematic model of their relationship and which fatty acid is more effective for WOCs and COCs or granulosa and cumulus cells.
Response: In our study, we used fatty acids because our mouse lines have very different FA profiles in all tissues (as visible on the heatmap from Calanni-Pileri et al., 2022, Int J Mol Sci, Figure 11A, Ref. #1, see below). FL1 mice show the largest differences in the FA profile (as visible on PCA from Calanni-Pileri et al., 2022, Int J Mol Sci, Fig. 11B, Ref. #1, see below). This is the reason why we focused on the fatty acids that showed a higher concentration in the COCs of the fertility lines (mostly “FL1-like profile”) to see if they could contribute to increasing the performance of maturation of the oocytes of the control line. Even though the results of the maturation were not statistically significantly higher for those FAs, probably because of the number of oocytes used, we obtained interesting increases using the long-chain SFAs. Unfortunately, we cannot conclude which FAs are most effective. We would like to continue our research trying those FAs in different combinations and concentrations. As mentioned before, we provided a scheme of the method/technique used for the experiment as an additional figure (novel Figure 4).
Minor points
- Line 32: Dose “ctrl” means control?
Response: We have two meanings of “control” in our manuscript. The first meaning is the "control line", i.e. the unselected control line that comes from the same founder population as the two fertility lines (information included in the Materials and Methods section, line 325-336). The second meaning is the "control condition", which corresponds to one of the combinations of the experimental design (COCs without FAs during maturation or nonsupplemented COCs, see Figure 3). In our IVM experiment, we only used oocytes from the “control line” and different treatments (oocytes with cumulus cells (COCs) or without cumulus cells (WOCs) with FAs or without (nonsupplemented) FAs). We changed “ctrl” to “control” in the manuscript, where the meaning is the “control line”. We used “ctrl” only on line 342 to identify the “ctrl condition” (nonsupplemented COCs).
- Lin57: Amend “dies” to “die”.
Response: We fixed the sentence as suggested.
- Line 102: Add a comma after “Currently”.
Response: We added the comma as requested.
- Line 152: Amend “C22.0” to “C22:0”.
Response: We swapped the period for the colon as suggested.
- In Figure 3, the maturation (%) results in the figure and table are not match.
Response: In Figure 3, the results on the figure are represented per small group of oocytes (the percentage of mature oocytes is calculated per day in the same group). The results in the table are represented as the total of the entire period in the same group (percentage of the total number of mature oocytes divided by the total number of rescued oocytes).
E.g., if on day1 we have 3/10 mature oocytes, on day2 we have 5/10 and on day3 2/10, on the graph we considered 30% on day1, 50% on day2 and 20% on day3, whereas in the table we considered the total of 10/30 (33%). We added this explanation in figure legend 3 (lines 217-220).
ARCOS, A., DE PAOLA, M., GIANETTI, D. & AL., E. 2017. α-SNAP is expressed in mouse ovarian granulosa cells and plays a key role in folliculogenesis and female fertility. Sci Rep, 7, 11765.
LANGHAMMER, M., WYTRWAT, E., MICHAELIS, M., SCHON, J., TUCHSCHERER, A., REINSCH, N. & WEITZEL, J. M. 2021. Two mouse lines selected for large litter size display different lifetime fecundities. Reproduction, 161, 721-730.
PORRAS-GÒMEZ, T. J. & MORENO-MENDOZA, N. 2020. Interaction between oocytes, cortical germ cells and granulosa cells of the mouse and bat, following the dissociation–re-aggregation of adult ovaries. Zygote, 1:10.
SCHÜLER, L. & BÜNGER, L. 1982. The Reproductive Lifetime Performance of Laboratory Mouse Lines Selected for Fertility. Archives of Animal Breeding 25, 275-281.

Round 2
Reviewer 3 Report
I think that the revised manuscript has been fundamentally improved and that it includes the contents requested by the referees and editorial team.